# Does the Recovery of Respiratory Viruses Impact Pulmonary Function at Baseline and 1-, 6-, and 12-Month Follow-Up in People Living with HIV and Pneumonia?

**DOI:** 10.3390/v16030344

**Published:** 2024-02-23

**Authors:** Iván Arturo Rodríguez-Sabogal, Ruth Cabrera, Diana Marin, Lucelly Lopez, Yudy Aguilar, Gustavo Gomez, Katherine Peña-Valencia, Will Riaño, Lázaro Vélez, Yoav Keynan, Zulma Vanessa Rueda

**Affiliations:** 1School of Medicine, Universidad de Antioquia, Medellin 050010, Colombia; ivan.rodriguezmd@gmail.com (I.A.R.-S.); anderson.riano@udea.edu.co (W.R.); lazaro.velez@udea.edu.co (L.V.); 2Infectious Diseases Section, Hospital Universitario San Vicente Fundación, Medellin 050010, Colombia; 3Escuela de Ciencias de la Salud, Facultad de Medicina, Universidad Pontificia Bolivariana, Medellin 050031, Colombia; ruth.cabrera@upb.edu.co (R.C.); dianamarcela.marin@upb.edu.co (D.M.); lucelly.lopez@upb.edu.co (L.L.); yudyagui@gmail.com (Y.A.); 4Grupo Biología de Sistemas, Facultad de Medicina, Universidad Pontificia Bolivariana, Medellin 050031, Colombia; 5Grupo de Investigación en Salud Pública, Facultad de Medicina, Universidad Pontificia Bolivariana, Medellin 050031, Colombia; katherine.penav@udea.edu.co; 6Grupo Investigador de Problemas en Enfermedades Infecciosas—GRIPE, Facultad de Medicina, Universidad de Antioquia, Medellin 050010, Colombia; 7Pulmonologist Section, Hospital Universitario San Vicente Fundacion, Medellin 050010, Colombia; gusgoco@hotmail.com; 8Escuela de Microbiología, Universidad de Antioquia, Medellín 050010, Colombia; 9Grupo Bacterias & Cancer, School of Medicine, Universidad de Antioquia, Medellin 050010, Colombia; 10Department of Medical Microbiology and Infectious Disease, University of Manitoba, Winnipeg, MB R3E 0J9, Canada; 11Department of Internal Medicine, University of Manitoba, Winnipeg, MB R3E 0J9, Canada; 12Department of Community Health Sciences, University of Manitoba, Winnipeg, MB R3E 0J9, Canada

**Keywords:** HIV, pneumonia, respiratory viruses, lung function, cohort study

## Abstract

The frequency of respiratory viruses in people living with HIV (PLHIV) and their impact on lung function remain unclear. We aimed to determine the frequency of respiratory viruses in bronchoalveolar lavage and induced sputum samples in PLHIV and correlate their presence with lung function. A prospective cohort of adults hospitalized in Medellín between September 2016 and December 2018 included three groups: group 1 = people diagnosed with HIV and a diagnosis of community-acquired pneumonia (CAP), group 2 = HIV, and group 3 = CAP. People were followed up with at months 1, 6, and 12. Clinical, microbiological, and spirometric data were collected. Respiratory viruses were detected by multiplex RT-PCR. Sixty-five patients were included. At least 1 respiratory virus was identified in 51.9%, 45.1%, and 57.1% of groups 1, 2 and 3, respectively. Among these, 89% of respiratory viruses were detected with another pathogen, mainly *Mycobacterium tuberculosis* (40.7%) and *Pneumocystis jirovecii* (22.2%). The most frequent respiratory virus was rhinovirus (24/65, 37%). On admission, 30.4% of group 1, 16.6% of group 2, and 50% of group 3 had airflow limitation, with alteration in forced expiratory volume at first second in both groups with pneumonia compared to HIV. Respiratory viruses are frequent in people diagnosed with HIV, generally coexisting with other pathogens. Pulmonary function on admission was affected in patients with pneumonia, improving significantly in the 1st, 6th, and 12th months after CAP onset.

## 1. Introduction

Respiratory diseases are the leading cause of morbidity and mortality in individuals with human immunodeficiency virus infection in both the pre- and post-HAART (highly active antiretroviral therapy) stage [1,2,3,4]. The overall rate of community-acquired pneumonia (CAP) in adults without human immunodeficiency virus (HIV) is as high as 11 cases per 1000 person-years (PY) and increases with age and the presence of comorbidities [5,6]. In individuals diagnosed with HIV, the risk of developing pneumonia is 25 times higher than in the general population, increasing with decreasing CD4 count [4,7].

Despite the development of new diagnostic methods and antimicrobial agents, CAP continues to be the leading cause of death of infectious origin in all age groups with and without HIV, with an estimated 3.1 million deaths worldwide in 2012, with mortality attributable to CAP ranging from 1 to 5% with outpatient treatment, 5 to 25% in hospital and up to 50% in ICU [7,8], and in 2011 represented costs of more than 10 billion dollars in the United States [7]. In Colombia, among the causes of hospitalization in people diagnosed with HIV, opportunistic infections are the leading category, with 23 to 30% attributed to tuberculosis, *Pneumocystis jirovecii* pneumonia accounting for 6 to 21%, and bacterial pneumonia accounting for 14% [8,9]. As in individuals without HIV, identifying the microorganisms involved in pulmonary infection allows for the establishment of measures such as providing adequate targeted treatment [10], de-escalation of broad spectrum antimicrobial therapy, more rational use of antibiotics [11], and implementing prophylaxis and vaccination measures. In individuals living with HIV, not identifying the etiology of pneumonia has been considered an independent mortality factor [10]. Unfortunately, despite the use of all available diagnostic tools, fewer than half of episodes receive a definitive etiological diagnosis, reaching up to 70% in the best-case scenario [4,9,12,13].

In people living with HIV who are diagnosed with bacterial pneumonia, the most frequently identified microorganisms are similar to those of individuals without HIV, first *Streptococcus pneumoniae* in 20 to 40%, followed by *Haemophilus Influenzae* (10 to 30%), *Staphylococcus aureus* and *Pseudomonas aeruginosa* (3 to 10%). Respiratory viruses (respiratory syncytial virus, parainfluenza virus, influenza virus, human metapneumovirus, adenovirus, coronavirus, and rhinovirus) have traditionally been considered infrequent in individuals living with HIV [14], probably due to the low sensitivity of serological tests in these patients, the difficulty or impossibility of culturing them, and the limited and non-standardized use of molecular biology techniques [4,9,15,16]. Studies using nucleic acid-based testing such as polymerase chain reaction (PCR) have improved the identification of respiratory viruses, making it possible to demonstrate the role of epidemic viruses such as influenza A H1N1, the presence of rhinoviruses, and the identification of others such as cytomegalovirus (CMV) and herpes simplex virus (HSV) [13,17]. However, their prevalence and role in individuals living with HIV, and the role they play in a normal virome, co-pathogens or principal causative pathogens, is still unclear.

HIV infection independently affects lung function. Several studies report that individuals living with HIV have a higher incidence of infectious and non-infectious lung diseases such as chronic obstructive pulmonary disease (COPD), pulmonary fibrosis, cancer, and pulmonary hypertension, with divergent results for asthma [1,18]. It is generally accepted that HIV infection is an independent risk factor for developing COPD with an odds ratio ranging from 1.47 to 10.93 after adjusting for other variables [19] such as smoking, cannabis or intravenous drug use, sex, age, and other pulmonary infections. More recent longitudinal cohort studies have demonstrated accelerated respiratory function decline and doubling of the incidence of COPD that correlated with unsuppressed viral replication and smoking [20,21]. The development of *P. jirovecii* pneumonia and CAP is considered an additive factor to lung function impairment, which has been demonstrated in retrospective studies, mostly without follow-up of pulmonary function tests [22]. There are no studies on the effect of atypical bacteria or respiratory viruses on lung function in individuals diagnosed with HIV.

We sought to determine the frequency of respiratory viruses in people diagnosed with HIV and pneumonia, and their effect on lung function. To achieve these objectives, we determined the frequency of respiratory viruses by real-time PCR (RT-PCR) in respiratory samples of bronchoalveolar lavage (BAL) and induced sputum from persons diagnosed with HIV and pneumonia, HIV, and pneumonia, and correlated the presence of these respiratory viruses with airflow limitation and the effect on spirometric variables assessing lung function at 1-, 6-, and 12-month follow-ups.

## 2. Materials and Methods

A prospective cohort study conducted between September 2016 and December 2018 included three groups: group 1 = people diagnosed with HIV and pneumonia, group 2 = HIV, and group 3 = pneumonia. Participants were recruited in two high-complexity institutions in Medellin, Colombia.

Individuals over 18 years of age who voluntarily agreed to participate in the study and who, according to the assigned group, were diagnosed with HIV or had a diagnosis of community-acquired pneumonia (CAP) were included. For this study, CAP was defined as acute lower respiratory tract infection associated with radiographic changes that were not explained by another condition, and excluding healthcare or ventilator-associated pneumonia. HIV infection was defined as those individuals with serologic demonstration of HIV by ELISA or positive viral load. Exclusion criteria were individuals with hospitalization in the previous two weeks, antibiotic use for more than 72 h in the last week, other causes of known immunosuppression other than HIV (use of prednisone ≥0.3 mg/kg/day for 3 weeks or more or ≥1 mg/kg/day for more than seven days, or its equivalent in other steroids; cytotoxic agents (except low-dose methotrexate: ≤15 mg/week), presence of hematologic malignancies, granulocytopenia <500 cells/mm^3^), obstructive cancer pneumonia, significant chronic lung disease (cystic fibrosis, severe COPD (forced expiratory volume at first second measured in liters-FEV1 < 50%), bronchiectasis and asthma), orotracheal intubation at study entry, major contraindication to induced sputum collection, inability to perform spirometry due to clinical condition, high probability of loss to follow-up (history of elopement or voluntary discharge, substance-related disorder classified as serious [23], severe mental disorder with progressive and marked deterioration in functioning) [24], not having stable or permanent housing, or living outside the metropolitan area of Medellin.

All hospitalized patients who met the inclusion criteria and who agreed to participate and signed the informed consent form were admitted consecutively. Variables were collected at the time of admission, including demographic information (sex, age, occupation, origin, socioeconomic stratum and health insurance [contributive or subsidized]), comorbidities (COPD, diabetes, renal failure, heart failure, hepatopathies, collagenopathies, neoplasms, pregnancy, hypo- and hypersplenism, epilepsy, altered consciousness or swallowing, previous pneumonia), clinical characteristics (respiratory symptoms, vital signs, weight, height, presence of lymphadenopathy, or organomegaly), vaccination status, previous antibiotic use, smoking, psychoactive substance or alcohol use, radiological and laboratory studies (CD4 count, HIV viral load, electrolytes, renal and liver function, arterial blood gases, and tests for diagnosis of other sexually transmitted and blood-borne infections such as syphilis and hepatitis B and C). The severity of pneumonia was assessed with the pneumonia severity index (PSI) and the CURB-65 [6,25].

During hospitalization, patients underwent routine diagnostic studies according to the concept of their treating physicians. In addition, venous blood and urine samples were collected at admission. Per protocol, all people living with HIV and with a diagnosis of pneumonia (group 1) that required hospitalization underwent BAL performed by the respiratory medicine group of each institution following a standardized protocol [9]. In all groups, we collected induced sputum (IS) samples if the person did not have a major contraindication (history of massive epistaxis necessitating an emergency room visit, history of bleeding disorders, history of heart failure, chest tube drainage for pneumothorax, recent eye surgery, and history of severe asthma requiring treatment in the intensive care unit) following previous research [26]. To detect respiratory viruses, RNAlater^®^ was added to each sample of BAL and IS, thus guaranteeing the stability of the RNA.

Conventional microbiological diagnostic studies were performed at the discretion of the treating physician and included direct studies of respiratory secretions in sputum and/or BAL (Gram, KOH, ZN, methenamine silver, toluidine blue, Wright) [9,27], cultures for aerobic bacteria, fungi and mycobacteria, and molecular tests for tuberculosis (GeneXpert^®^ or Anyplex^®^). In addition, treating physicians ordered blood cultures, stains and cultures in pleural fluid, lung biopsy, pleura and lymph nodes, PCR for tuberculosis in urine, and urinary antigen for *Histoplasma capsulatum* in some patients based on their clinical history and findings. All people received treatment accordingly, and the antibiotics and other treatments were decided by the treating physician(s).

Likewise, respiratory viruses were identified in BAL and IS samples by molecular tests. RNA/DNA extraction was performed with the commercial kit Ribospin^TM^ vRD (GeneAll Biotechnology Co., Ltd., Seoul, Repulic of Korea) according to the manufacturer’s recommendations. Prior to extraction, the RNAlater^®^ present in the sample was removed by adding the same amount of cold PBS and centrifuging at 5000× *g*. If remnants were still observed, a second wash was performed. In each extraction run, a negative control was added as a quality control. A multiplex one-step RT-PCR Allplex™ Respiratory Panel 1, 2, and 3 (Seegene^®^, Seoul, Repulic of Korea) was used to detect respiratory viruses in BAL and IS samples. Allplex™ Respiratory Panel 1 detected influenza A virus (Flu A), influenza A-H1 (Flu A-H1), influenza A-H1pdm09 (Flu A-H1pdm09), influenza A-H3 (Flu A-H3), influenza B virus (Flu B), respiratory syncytial virus A (RSV A), respiratory syncytial virus B (RSV B), and internal control (IC). Allplex™ Respiratory Panel 2 detected adenovirus (AdV), enterovirus (HEV), metapneumovirus (MPV), parainfluenza virus 1 (PIV 1), parainfluenza virus 2 (PIV 2), parainfluenza virus 3 (PIV 3), parainfluenza virus 4 (PIV 4), and internal control (IC). Allplex™ Respiratory Panel 3 detected bocavirus 1/2/3/4 (HBoV), coronavirus 229E (229E), coronavirus NL63 (NL63), coronavirus OC43 (OC43), human rhinovirus (HRV), and internal control (IC).

Each participant underwent simple spirometry at baseline and 1, 6, and 12 months using the Maryland, U.S. FDA (Food and Drug Administration)-approved, EasyOne^TM^ ultrasonic spirometer (ndd Medical Technologies, Zurich, Switzerland) in accordance with the guidelines of the American Thoracic Society and the European Respiratory Society [28], ensuring measurements of forced expiratory volume at first second (FEV1), forced vital capacity (FVC), and FEV1/FVC ratio of acceptable quality, from which the best of three tests were selected for analysis [29]. The presence of airflow limitation was established according to the recommendations of the Global Lung Function Initiative (GOLD) guidelines using FEV1/FVC < lower limit of the normal range (LLN) and FEV1/FVC < 0.70, standardized for Latin population [30]. An individual was considered to have airflow limitation when the person had an FEV1/CVF ratio lower than the LLN; this LLN is equivalent to the 5th percentile of the reference population (healthy and non-smoking) standardized by age (equivalent to minus 1.64 z-score). The other definition accepted by these guidelines is the FEV1/FVC ratio < 0.70; however this option tends to underestimate the presence of COPD in young individuals and overestimate it in those older than 45 years, which should be taken into account for the analysis of people diagnosed with HIV who are generally young.

In individuals who met the criteria for airflow limitation, severity was quantified according to the percentage of FEV1% predicted. Severity assessment was defined as the percentage of FEV1 (L) of predicted, and defined as follows: mild ≥ 80%, moderate between 50 to 79%, severe between 30 to 49% and very severe <30%. In addition, independent FEV1 and FVC values and variations were analyzed to identify changes below the definition of airflow limitation (COPD), correlating these results with sex, age, body mass index (BMI), previous pulmonary infection, HIV infection, cigarette smoking, CD4 count, HIV viral load, and presence or absence of respiratory viruses.

### Statistical Analysis

We designed a database in Microsoft Access^®^ version 2019, which was used to record sociodemographic, clinical, and microbiological characteristics, including the respiratory viruses identified. We performed descriptive statistics estimating measures of total and relative frequency, and median and interquartile range (IQR, percentile 25–75) because the quantitative variables did not have a normal distribution. The Chi-square test was used to analyze the differences between groups in terms of the presence of airflow limitation (FEV1/FVC < LLN or 0.70) both at admission and at the 1-, 6-, and 12-month follow-ups. In addition, variation in spirometric measures (FEV1, FVC, in liters) and FEV1/FVC ratio were analyzed, using the Friedman test, for minimal changes that did not meet the criteria for airflow limitation. In all comparisons, a *p*-value < 0.05 was considered significant, and the analyses were performed in IBM SPSS^®^ version 24.0 and Stata^®^ version 14.1.

## 3. Results

Two hundred and forty-eight (248) people were screened consecutively (Figure 1), of which 65 were included, forming the groups of HIV and pneumonia (n = 27; group 1), HIV (n = 31; group 2), and pneumonia (n = 7; group 3). The main reasons for exclusion were high risk of loss to follow-up, antibiotic use for more than 72 h, and living outside the metropolitan area (Appendix A describes the causes of exclusion).

Of all those included, 71% were men with a median age of 39 years (IQR 27–51); 14% were smokers; 30% had received vaccination against pneumococcus and influenza; 22% had some comorbidity (COPD, diabetes, renal or cardiac failure or chronic liver disease). The body mass index (BMI) of individuals in the HIV group was lower than in those in the other groups (20.3; IQR 16.7–23.3). The pneumonia group had higher antibiotic use in the last 3 months. In individuals diagnosed with HIV, the HIV/pneumonia group had the lowest CD4 count (median 56 vs. 414 cells/µL), and only 38% of this group received antiretrovirals vs. 73% in the HIV group (Table 1).

There were no differences in respiratory symptoms or clinical findings between groups with CAP or in individuals with respiratory viruses. We also found no significant differences in pneumonia severity scales (PSI and CURB65), smoking exposure, previous pneumonia, vaccination coverage, and trimethoprim sulfamethoxazole prophylaxis. An important finding was the low vaccination rate found in all three groups (without serologic confirmation or other immunization record), with vaccination against pneumococcus in 25%, 35%, and 14%; and against influenza in 22%, 41% and 14% in the HIV/pneumonia, HIV, and pneumonia groups, respectively, suggesting problems in the implementation of current immunization recommendations. Mortality at hospital discharge occurred in 6 patients, of which four were in group 1 (HIV/pneumonia group), one in group 2 (HIV), and one in group 3 (pneumonia).

### 3.1. Microbiological Findings

In patients with CAP, at least one microorganism (aerobic bacteria, mycobacteria, fungi, or viruses) was identified in 87.5% (24/27) of the HIV/pneumonia group, and in 57.1% (4/7) of the pneumonia group. No studies were performed for atypical bacteria or anaerobes.

In the HIV/pneumonia group, the most common microorganisms were *M. tuberculosis* 11/27 (40.7%), followed by *P. jirovecii* 6/27 (22.2%), Gram-negative bacilli 4/27 (14.8%), *S. pneumoniae* 3/27 (11.1%), S. *aureus* 2/27 (7.4%), and *H. influenzae* 1/27 (3.7%). In the pneumonia group, there was only one identification that was polymicrobial (more than 3 organisms determined to likely represent contamination).

We found at least 1 respiratory virus in 49.2% of all patients, with no significant differences between groups. In 5 cases, we identified 2 respiratory viruses simultaneously in the same patient. In the HIV/pneumonia group, in 2 cases, only respiratory viruses were identified as a possible etiology of pneumonia. In the pneumonia group, there were no microbiological findings other than respiratory viruses (4/7 cases). The most frequently identified respiratory viruses were rhinoviruses, with no significant difference in the 3 groups, followed by parainfluenza 1 (PIV1), parainfluenza 3 (PIV3), influenza A (FluA), metapneumovirus (MPV), parainfluenza 4 (PIV4), influenza B (FluB), coronavirus NL63, respiratory syncytial virus A (RSV A) and adenovirus; we did not detect any cases of parainfluenza 2 (Table 2).

Eighty-nine percent of respiratory viruses were associated with another respiratory pathogen (*M. tuberculosis*, *P. jirovecii,* aerobic bacteria); respiratory viruses were found without another microorganism in only 4 cases (3 of rhinovirus and 1 of parainfluenza 3) (Appendix A).

In the HIV group, a wide variety of conditions led to hospitalization. The most frequent were syphilis, acute diarrhea, headache of unclear cause, heart failure, nodal tuberculosis, intestinal tuberculosis, cerebral toxoplasmosis, and *Paracoccidioides brasiliensis* (Appendix A).

### 3.2. Lung Function Studies-Spirometric Findings

There were four deaths in group 1, one in group 2, and one in group 3 during hospitalization; therefore, spirometry data were obtained at admission in 23 individuals in group 1 (HIV/pneumonia group), 30 in group 2 (HIV), and 6 in group 3 (pneumonia). At follow-up, there were two withdrawals in group 2 and one in group 3, and there were additional deaths and loss to follow-up. 

Using the definition of airflow limitation based on FEV1/FVC < LLN, 7 patients (30.4%) in the HIV/pneumonia group, 5 (16.6%) in the HIV group, and 3 (50%) in the pneumonia group were identified in the admission spirometry. Follow-up spirometry at 1, 6, and 12 months showed a decrease in the number of cases of airflow limitation in the three groups, demonstrating the improvement in spirometric variables over time. Using the FEV1/FVC < 0.70 definition, fewer patients with airflow limitation were identified, suggesting differences in applying the two definitions to diagnose COPD (Table 3). 

In the subgroup of patients who at admission had FEV1/FVC < LLN meeting the definitions of COPD, an analysis was performed to evaluate severity, finding most patients in the moderate category; see Appendix A. 

In the pneumonia groups (with and without HIV), the effect of the acute condition (pneumonia) on FEV1 was evident. On admission, spirometry in the HIV/pneumonia group had a median FEV1 of 1.80 L (IQR 1.03–2.78), and in the pneumonia group, a median of 1.38 L (IQR 0.70–1.68) was found, which contrasts with the median FEV1 of the HIV group, which was 3.21 L (IQR 2.46–3.82) (Table 4). 

Spirometric follow-up in the pneumonia groups (with and without HIV) showed a progressive increase in FEV1, probably due to the expected recovery of lung function after resolution of acute pulmonary infection, which was strikingly faster in individuals with HIV and CAP. The HIV group had a higher FEV1 than the other groups from admission and remained without significant changes over time; similar findings were found when analyzing FVC and FEV1/FVC ratio (Table 4 and Figure 2). 

Analysis of FEV1 and FVC as a percentage of predicted both at admission and at 1-, 6-, and 12-month follow-ups showed similar results (Appendix A).

When correlating the presence of respiratory viruses in the three study groups with the variables FEV1, predicted FEV1, FVC, and predicted FVC, at admission and in the follow-up spirometries, there were no differences between the groups. However, the FEV1/FVC ratios in individuals with respiratory viruses revealed lower values in the three groups without being statistically significant, maintaining the previously mentioned trends of improvement over time (Table 5).

The type of microorganism identified (mycobacteria, fungi, bacteria, or virus) in any of the groups was not associated with particular patterns of spirometry results. Patients with a history of previous pneumonia in all groups had lower FEV1 values on admission spirometry. No significant differences were found between spirometry results with age, sex, smoking exposure, CD4 count, or HIV viral load. 

## 4. Discussion

This study demonstrated the presence of respiratory viruses in similar proportions in individuals with HIV with or without pneumonia and those without HIV with pneumonia. In addition, it showed the reduction of pulmonary function evaluated by spirometry after acute pneumonia and recovery therefrom during follow-up.

The study is notable for the high percentage of microorganisms identified in the group with HIV and pneumonia. We found at least one respiratory virus in 51.9%, 45.1%, and 57.1% of the patients in the HIV/pneumonia, HIV, and pneumonia groups, respectively. One of the few and most recent studies of etiologic investigation of pneumonia in people with HIV demonstrated the presence of respiratory viruses in 6% of patients using traditional techniques (serology, cultures), which increased to 21% when molecular biology testing (PCR) was used [13]. A similar finding was reported by Garbino et al. in BAL samples, finding at least one respiratory virus in 18.6% of people diagnosed with HIV with respiratory symptoms [31]. In Spain, another study of individuals with HIV and pneumonia detected the presence of respiratory viruses in 20.8% of patients using serological tests, cultures and PCR for adenovirus, rhinovirus, influenza, enterovirus, respiratory syncytial virus, and coronavirus in nasopharyngeal swab samples [16]. Respiratory viruses have also been reported in children with HIV and lower respiratory symptoms in Africa at higher frequencies, with at least one respiratory virus in 53% of children according to PCR on nasal swabs, a finding that may be explained in part by colonization of the upper airway [32]. Recently, Maartens et al. in Cape Town, South Africa prospectively studied 284 patients with HIV and lower respiratory infections, in whom they applied multiplex PCR to induced sputum samples, finding a prevalence of pneumonia of 52%, of PjP of 34%, and of tuberculosis of 41%. Respiratory viruses were detected in 71% of all patients without establishing their pathogenic role. Coinfections were common and diverse [17]. On the other hand, in patients with pneumonia without HIV, the identification of respiratory viruses is very variable, reported between 15% to 37% in different studies [33]; moreover, their pathogenic role is not conclusive, either. One explanation for the higher proportion of respiratory viruses found in our study may be the use of multiplex real-time PCR with the capacity to identify 16 different viruses. 

The main virus we Identified was rhinovirus in 37% of HIV/pneumonia patients, 35% of HIV patients, and 42% of pneumonia patients, similar to what has been found in other studies [13,16,32,34]. Another important aspect was the co-infection of respiratory viruses with other viruses, *M. tuberculosis*, *P. jirovecii*, and bacteria (nearly 90%), a similar finding reported in other publications, such as the study by Camps et al., where three etiological categories were defined, one of which included the combination of virus with other agent [34] and the aforementioned study by Maartens [17]. In our study, the most common combination was a respiratory virus with *M. tuberculosis*, followed by respiratory viruses and *P. jirovecii.*

Multiple hypotheses can explain the effect of HIV and the presence of other microorganisms on the progression of airflow limitation, which can be clinically translated into COPD. The main factor appears to be systemic immune activation in which the CD8+ T cell and macrophage response increases generating chronic lymphocytic alveolitis with release of proteases and inflammatory cytokines (IFNγ) that cause lung damage. There is an imbalance with decreased antioxidant levels and increased oxidants that directly damage the lung parenchyma, and there is increased susceptibility to apoptosis stimulated in endothelial cells by HIV Tat and Naf proteins. Additionally, pulmonary infections such as tuberculosis, *P. jirovecii* pneumonia, and bacterial pneumonia promote the inflammatory response and lung damage [35], which can generate a further reduction in lung function, between 109 mL to 264 mL of FEV1 and 117 mL to 254 mL of FVC [36,37]. Other factors with a possible effect on lung function impairment are colonization of the airway by other microorganisms, such as atypical bacteria and changes in the pulmonary microbiome [19]. A cohort study following 265 individuals for a median of 8.1 years used a data-driven model to describe longitudinal pulmonary function phenotypes among PLHIV. They found that current smoking and past years of smoking were predictors of adverse FEV_1_ and FEV_1_/FVC, and that detectable viremia was the only HIV infection marker associated with the adverse Dl_CO_ phenotype. The authors identified two trajectories each for FEV_1_ and FVC, characterized by a faster rate of decline in participants with higher baseline lung function. These findings are consistent with observations in population-based and COPD cohorts. They also found a correlation between CRP and endothelin-1 levels, representing inflammation and endothelial dysfunction, respectively, with adverse FEV and FVC phenotypes [38].

In the pneumonia groups (with and without HIV), the initial FEV1 was lower than in the group without pneumonia, suggesting that the acute infectious event is directly related to the deterioration of pulmonary tests, beyond the presence of HIV or respiratory viruses. The low lung volumes at the 6-month follow-ups suggest underlying lung pathology that may have predisposed to the infection to playing a role in the persistent dysfunction. In these groups with pneumonia, progressive improvement of pulmonary function was documented at follow-up. Our limited duration of 1-year follow-up did not allow for assessment of the trajectories of functional decline. There are few studies of pulmonary function in HIV; one of them evaluated the prevalence of airflow limitation at a single moment in patients with HIV without pneumonia and healthy controls without HIV, finding 10.6% in both populations, with a slight difference in FEV1, which was lower in patients with HIV [39]. Other studies have shown decreased pulmonary function in individuals with HIV after presenting pulmonary infection by *P. jirovecii* and bacterial pneumonia, using spirometry and diffusing capacity for carbon monoxide (DLCO) [40,41,42]. The ALIVE study, in which spirometry (FEV1, FVC, FEV1/FVC) was performed after adjustment for clinical variables (sex, age), behavioral variables (smoking), and some infections, showed that uncontrolled HIV infection was independently associated with lung function impairment. Although not statistically significant, the annual rate of reduction in forced expiratory volume at first second (FEV1) in individuals with HIV was −35.8 mL/year (95% CI −51.2 to −20.3 mL/year) compared to −23.6 mL/year (95% CI −32.6 to −14.7 mL/year) among those not infected with HIV (*p* = 0.135); forced vital capacity (FVC) in people living with HIV decreased −9.29 mL/year (95% CI −25.1 to 6.5 mL/year), and in those not infected with HIV, it was +8.0 mL/year (95% CI −26.4 to 18.7 mL/year) (*p* = 0.05) [36]. When researchers stratified patients, the patients evidenced greater pulmonary deterioration in individuals with HIV viral load (CV-HIV) greater than 75,000 copies/mL and CD4 <100 cells/mm^3^, with significant reduction in FEV1 and FVC in individuals with HIV vs. those without HIV of −99.1 mL/year vs. −23.5 mL/year (*p* = 0.004) for FEV1 and −74.0 mL/year vs. +8.24 mL/year in FVC (*p* = 0.008). Varkila et al. studied individuals with HIV by post-bronchodilator spirometry vs. individuals without HIV, demonstrating greater deterioration of lung function in individuals diagnosed with HIV with independent effect on outcome in those with tuberculosis [43]. In our study we found no significant differences between pulmonary function and CD4 count or HIV viral load. Spirometric follow-up needs to be prolonged to find possible differences, perhaps at least 5 years. The subgroup of patients in the START study, in which spirometric follow-up was performed at 3.9 years, showed no significant differences [44].

The main limitations of this study are as follows: 1. Due to lack of resources, it was not possible to perform all the microbiological diagnostic tests to include identification by utilizing bacterial and *H. capsulatum* urinary antigens, which could have increased the proportion of identified microorganisms. 2. As in other studies, defining whether the presence of respiratory viruses corresponds to asymptomatic infection (“colonization”) is complex and will continue to be a challenge that should be evaluated in future studies, perhaps by analyzing the quantitative values of RT-PCR and using repeated measures to evaluate respiratory virus shedding after an acute infection, in particular among people with immunosuppressive conditions. Previous research has shown that certain conditions, such as older age, multiple comorbidities, solid organ transplant may delay PCR clearance, but not by degree of immunosuppression [45]; however, other studies have shown long-term shedding of influenza, parainfluenza, and respiratory syncytial viruses in people with hematological disorders [46]. 3. As for pulmonary function tests, DLCO measurement was not performed, which allows for a dynamic evaluation of pulmonary function, although it represents greater expenses and operational difficulties. 4. Finally, our short follow-up time does not allow to evaluate long-term lung dysfunction. An extended follow-up, of 2 to 5 years, would allow for changes in pulmonary function described in individuals post recovery from pneumonia to be observed. 5. The small sample size, in particular in group 3 (pneumonia), is exploratory. Therefore, our results are limited to people who have similar conditions to the participants in each group.

In conclusion, our findings demonstrate that respiratory viruses are frequent in people diagnosed with HIV, generally coexisting with other pathogens. We could not identify any difference in lung function or clinical outcomes among those coinfected with respiratory viruses. The role respiratory viruses play in lung normal microbiome or as contributors to the pathogenesis of lung infection and dysfunction in people living with HIV remain to be elucidated. In patients with CAP, pulmonary function was affected on admission and improved significantly in the first months after acute infection (CAP). Further studies with a larger number of patients and prolonged follow-up are required. 

## Figures and Tables

**Figure 1 viruses-16-00344-f001:**
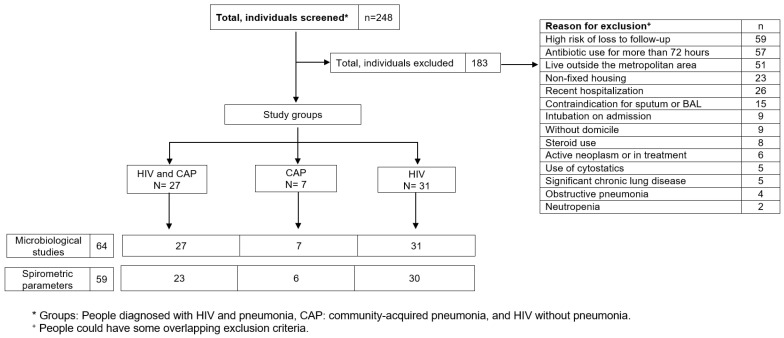
Patient recruitment and follow-up flowchart.

**Figure 2 viruses-16-00344-f002:**
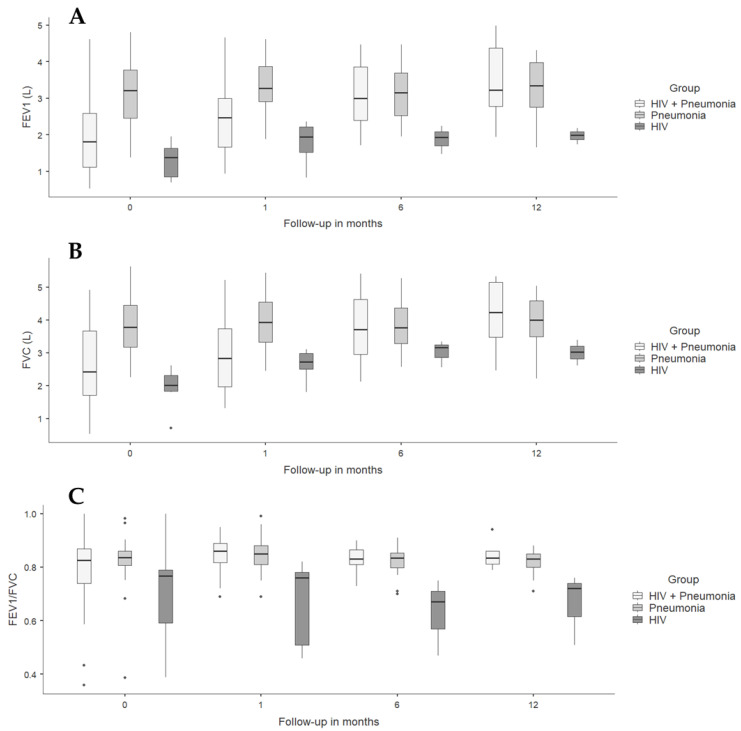
Spirometric variables in people diagnosed with HIV and/or pneumonia at admission, and at 1- and 6-month follow-ups. (**A**) FEV1 L: forced expiratory volume at first second measured in liters. (**B**) FVC: forced vital capacity. (**C**) FEV1/FVC.

**Table 1 viruses-16-00344-t001:** Baseline characteristics of people diagnosed with HIV and/or pneumonia in Colombia.

Variables	Group			*p*-Value
	HIV and Pneumonia n = 27	HIV	Pneumonia	
		n = 31	n = 7	
Male sex, n (%)	17 (63.0)	24 (77.4)	5 (71.4)	0.482
Age in years, median (IQR 25–75)	40 (27–51)	34 (26–44)	66 (53–75)	0.005
BMI in kg/m^2^, median (IQR 25–75)	20.3 (16.7–23.3)	23.4 (21.9–25.5)	24.3 (21.8–25.1)	<0.001
Pre-existing conditions, n (%)				
Antibiotic use in the last 3 months	4 (14.8)	5 (16.1)	4 (57.1)	0.034
Pneumococcal vaccination	7 (25.9)	11 (35.5)	1 (14.3)	0.414
Influenza vaccination	6 (22.2)	13 (41.9)	1 (14.3)	0.118
Prior diagnosis of pneumonia	6 (22.2)	4 (12.9)	1 (14.3)	0.628
Smoked in the last 6 months	9 (33.3)	10 (32.3)	2 (28.6)	0.972
COPD	0 (0.0)	0 (0.0)	3 (42.9)	0
Diabetes	1 (3.7)	1 (3.2)	0 (0.0)	0.878
Chronic renal failure	0 (0.0)	3 (9.7)	1 (14.3)	0.198
Congestive heart failure	0 (0.0)	2 (6.5)	2 (28.6)	0.02
Chronic liver disease	1 (3.7)	1 (3.3)	0 (0.0)	0.878
CD4 count, median (IQR 25–75)	56 (24–253)	414 (140–695)	NA	<0.001
<200 cells/µL	16 (66.7)	9 (30.0)		
200–500 cells/µL	7 (29.2)	7 (23.3)		
>500 cells/µL	1 (4.2)	14 (46.7)		
Viral load of HIV copies/µL, median (IQR 25–75)	71,477 (4710–255,764)	290 (41–196,155)	NA	0.061
≤100,000 copies/µL	14 (56.0)	16 (69.6)		
>100,000 copies/µL	11 (44.0)	7 (30.4)		
Time in months since diagnosis of HIV, median (IQR)	23 (1–117)	25 (4–133)	NA	0.011
Antiretroviral therapy, n (%)	10 (38.5)	22 (73.3)	NA	0.009

BMI: body mass index. IQR: interquartile range, COPD: chronic obstructive pulmonary disease.

**Table 2 viruses-16-00344-t002:** Detection of respiratory viruses in people diagnosed with HIV and/or pneumonia.

Detection of Respiratory Viruses *	Group		
	HIV/Pneumonia N = 27, n/N (%)	HIV N = 31, n/N (%)	Pneumonia N = 7, n/N (%)
At least 1 respiratory virus per patient	14/27 (51.9)	14/31 (45.1)	4/7 (57.1)
2 respiratory viruses in the same patient	2/27 (7.4)	2/31 (6.5)	1/7 (14.3)
Respiratory viruses, n (%)			
Rhinovirus	10/27 (37)	11/31 (35.5)	3/7 (42.9)
Parainfluenza 1	2/27 (7.4)		
Parainfluenza 3		1/31 (3.2)	1/7 (14.3)
Parainfluenza 4		1/31 (3.2)	
Influenza A	1/27 (3.7)		1/7 (14.3)
Influenza B		1/31 (3.2)	
Coronavirus 63	1/27 (3.7)		
Metapneumovirus	1/27 (3.7)	1/31 (3.2)	
Respiratory Syncytial Virus A		1/31 (3.2)	
Adenovirus	1/27 (3.7)		

* Studies performed on bronchoalveolar lavage and induced sputum samples.

**Table 3 viruses-16-00344-t003:** Airflow limitation according to spirometric parameters at admission and at 1, 6, and 12 months in people living with HIV and/or pneumonia.

Spirometry	HIV/Pneumonia n (%) *	HIV n (%) *	Pneumonia n (%) *	Total	*p*-Value
FEV1/FVC < LLN					
Baseline	7/23 (30.4)	5/30 (16.6)	3/6 (50)	15/59 (25.4)	0.180
1 month	2/18 (11.1)	1/21 (4.8)	2/5 (40)	5/44 (11.4)	0.148
6 months	1/14 (7.1)	2/16 (12.5)	1/3 (33.3)	5/33 (15.2)	0.082
12 months	1/10 (10.0)	1/17 (5.9)	2/3 (66.7)	4/30 (13.3)	0.060
FEV1/FVC < 0.70					
Baseline	4/23 (17.4)	2/30 (6.7)	2/6 (33.3)	8/59 (13.6)	0.197
1 month	1/18 (5.6)	1/21 (4.8)	2/5 (40)	4/44 (9.1)	0.116
6 months	0/14 (0.0)	0/16 (0.0)	2/3 (66.7)	2/33 (6.1)	0.004
12 months	0/10 (0.0)	0/17 (0.0)	1/3 (33.3)	1/30 (3.3)	0.084

* Number of patients with airflow limitation/number of patients studied with spirometry (%). FEV1: forced expiratory volume at first second, FVC: forced vital capacity, LLN: lower limit of the normal range (LLN).

**Table 4 viruses-16-00344-t004:** Comparison of FEV1 in people diagnosed with HIV and/or pneumonia, measured at admission and at months 1, 6, and 12.

Spirometry	Group 1 HIV/Pneumonia			Group 2 HIV			Group 3 Pneumonia		
	n	FEV1 L Median (IQR) *	*p*-Value	n	FEV1 L Median (IQR)	*p*-Value	n	FEV1 L Median (IQR)	*p*-Value
Baseline	23	1.80 (1.03–2.78)	0.003	30	3.21 (2.46–3.82)	0.774	6	1.38 (0.70–1.68)	0.086
1 month	18	2.47 (1.65–3.08)		21	3.27 (2.91–3.87)		5	1.94 (1.52–2.22)	
6 months	14	3.00 (2.27–3.90)		16	3.15 (2.52–3.77)		3	1.93 (1.48–2.24)	
12 months	10	3.23 (2.75–4.43)		17	3.34 (2.76–3.98)		3	1.99 (1.74–2.18)	

* Median forced expiratory volume at first second measured in liters (FEV1 L). IQR: interquartile range.

**Table 5 viruses-16-00344-t005:** Pulmonary function tests according to the presence of respiratory viruses at admission and at months 1, 6, and 12 in people diagnosed with HIV and/or pneumonia.

Spirometric Parameter		HIV/Pneumonia Median (IQR)		HIV Median (IQR)		Pneumonia Median (IQR)	
		Identification of Respiratory Viruses					
		Yes	No	Yes	No	Yes	No
FEV1/FVC	Baseline	0.79 (0.71–0.87)	0.83 (0.74–0.89)	0.83 (0.76–0.86)	0.85 (0.82–0.87)	0.75 (0.54–0.79)	0.79 (0.39–1.0)
	1 month	0.85 (0.81–0.89)	0.87 (0.85–0.90)	0.85 (0.77–0.88)	0.85 (0.82–0.88)	0.63 (0.48–0.79)	0.78 (0.78–0.78)
	6 months	0.84 (0.83–0.87)	0.82 (0.81–0.85)	0.82 (0.77–0.86)	0.84 (0.82–0.85)	0.57 (0.47–0.67)	0.75 (0.75–0.75)
	12 months	0.84 (0.83–0.86)	0.82 (0.80–0.86)	0.83 (0.75–0.84)	0.84 (0.82–0.88)	0.62 (0.51–0.72)	0.76 (0.76–0.76)
FEV1 L	Baseline	1.87 (0.93–3.33)	1.69 (1.11–2.15)	2.86 (2.03–3.94)	3.26 (2.89–3.58)	1.49 (1.28–1.95)	0.70 (0.70–1.68)
	1 month	2.11 (1.65–2.77)	2.58 (1.74–3.69)	3.49 (2.51–4.09)	3.23 (2.93–3.80)	1.87 (1.17–2.29)	1.94 (1.94–1.94)
	6 months	3.13 (2.89–3.90)	2.88 (2.18–4.01)	2.54 (2.35–3.61)	3.18 (3.11–3.98)	1.89 (1.48–2.24)	1.93 (1.93–1.93)
	12 months	3.8 (3.14–4.37)	2.91 (2.60–4.43)	3.21 (1.99–3.98)	3.34 (3.19–4.02)	1.96 (1.74–2.18)	1.99 (1.99–1.99)

FEV1 L: forced expiratory volume at first second measured in liters. FVC: forced vital capacity. IQR: interquartile range.

## Data Availability

The data presented in this study are available on request from the corresponding author. The data are not publicly available due to ethical reasons because at the time people were included in the study we did not request their permission to share their data publicly.

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
