# Peer review of "Does the Recovery of Respiratory Viruses Impact Pulmonary Function at Baseline and 1-, 6-, and 12-Month Follow-Up in People Living with HIV and Pneumonia?"

_viruses, 2024, doi:10.3390/v16030344_

Round 1

Reviewer 1 Report

Comments and Suggestions for Authors

Rodriguez-Sabogal and colleagues present data on the recovery of respiratory viruses from 65 hospitalized adult patients divided into three groups (HIV with pneumonia, HIV without pneumonia, and pneumonia without HIV). The biggest limitation of the study is the small size of the study cohort with smaller numbers in each subgroup, especially those with pneumonia with no HIV (n=7). The patients in each group had many serious co-morbidities which makes it hard to assess the effects of recovered viruses on the overall illness effects. In fact, when they compared the changes in spirometric parameters between those from whom a virus was recovered and those from whom no virus was recovered, the improvement over time was comparable. The title of the paper promises more than it delivers, as it suggests that viruses had an actual long-term impact, a point they did not establish. Perhaps a better title should read like this: Does the recovery of respiratory viruses impact pulmonary function at 6 months in people with HIV and pneumonia?

A few other points should be addressed in any future revision of this manuscript:

1. The Abstract needs a better description of the 3 groups of patients in the study. The way they describe them is confusing. Also, the last line of the Abstract mentions follow-up only at one month, while the title mentions six months.

2. Lines 129-142 of the Materials and Methods section describe a number of conditions that I assume are exclusion criteria, but the authors forgot to mention that.

3. Lines 161-165 describe the methods of collection of respiratory secretions. The pneumonia with HIV group underwent bronchoalveolar lavage (BAL) while those with HIV but no pneumonia and others with pneumonia but no HIV underwent induced sputum sampling. The collection methods have different sensitivities, with higher rates of recovery of potential pathogens using BAL sampling. In addition, there is no mention of attempts at respiratory virus sampling of the upper airway of HIV patients with pneumonia to assess whether these viruses could have been carried into the lower airways during the process of introducing the bronchoscopes.

4. There should be some mention of the indications for BAL in HIV patients with pneumonia at the authors' various medical centers. Do they perform BALs on all such patients? If not, this suggests that these are sicker HIV patients with pneumonia as compared to well-controlled HIV patients with a common community-acquired pneumonia (CAP). Looking at Table 1, these patients (HIV with pneumonia) have a much lower median CD4+ cell count, higher HIV RNA viral loads, and fewer were on antiretroviral therapy.

5. The 3 subgroups of patients in this study are very different from each other in many ways (e.g., age, COPD, congestive heart failure). The comparison between them is only useful if one concludes that the recovery of respiratory viruses from any of these groups has the same lack of short or long-term impact on their illnesses.

6. When was this study conducted? There is no mention of that. Curiously, looking at lines 175-187, there is no mention of looking for SARS-CoV-2 RNA.

7. The authors should mention that, while the multiplex PCR assay utilized in this study is quite broad, it does not cover all conceivable viral causes of respiratory tract infections. Hence, the conclusions are limited to the detectable viruses.

8. Under Results, Figure 1, the flow chart should contain the number of patients in each group (HIV and CAP, HIV, and CAP).

9. Line 239: The authors mention influenza and pneumococcal vaccine rates of uptake in the various groups, but there is no mention of any COVID-19 vaccine rates. This suggests that the study is not recent, when one takes into account the apparent lack of SARS-CoV-2 RNA testing. If so, the relevance of their data to current disease epidemiology in the various study subgroups diminishes considerably.

10. In Table 3, at least half the patients in each subgroup were lost to follow up. The authors should separately analyze changes in spirometric findings among those with testing at 0, 1, and 6 months for some homogeneity.

11. For the patients with TB, were they treated? Given the frequency of this diagnosis, perhaps an analysis of this group of patients is also warranted, since that (i.e., TB treatment) can account for observed improvement.

12. The authors should mention somewhere in the paper that patients with immune deficiency are likely to shed viruses for longer periods than immunocompetent patients. Since the PCR test is qualitative (present vs absent), the detection of any one virus does not tell us when the infection actually occurred (now or a few weeks before). Quantitative PCR assays can help, as the authors allude to in the text. 

Comments on the Quality of English Language

Good overall.

Author Response

Dear reviewer,

We want to thank you because thanks to your comments we realized that we were missing some important information, and some sentences or paragraphs were not detailed or were confusing. We incorporated the suggested changes using track changes to facilitate the review.

Rodriguez-Sabogal and colleagues present data on the recovery of respiratory viruses from 65 hospitalized adult patients divided into three groups (HIV with pneumonia, HIV without pneumonia, and pneumonia without HIV). The biggest limitation of the study is the small size of the study cohort with smaller numbers in each subgroup, especially those with pneumonia with no HIV (n=7). The patients in each group had many serious co-morbidities which makes it hard to assess the effects of recovered viruses on the overall illness effects. In fact, when they compared the changes in spirometric parameters between those from whom a virus was recovered and those from whom no virus was recovered, the improvement over time was comparable. The title of the paper promises more than it delivers, as it suggests that viruses had an actual long-term impact, a point they did not establish. Perhaps a better title should read like this: Does the recovery of respiratory viruses impact pulmonary function at 6 months in people with HIV and pneumonia?

Answer: Thanks. We changed the title as suggested by the reviewer.

A few other points should be addressed in any future revision of this manuscript:

  1. The Abstract needs a better description of the 3 groups of patients in the study. The way they describe them is confusing. Also, the last line of the Abstract mentions follow-up only at one month, while the title mentions six months.

Answer: Thanks for your comments. We reviewed the paper and the abstract entirely to make the paper clearer, clarified each group, and added the follow-up at months one, six, and 12.

  1. Lines 129-142 of the Materials and Methods section describe a number of conditions that I assume are exclusion criteria, but the authors forgot to mention that.

Answer: Yes, our apologies for missing these words, those were exclusion criteria. We corrected.

  1. Lines 161-165 describe the methods of collection of respiratory secretions. The pneumonia with HIV group underwent bronchoalveolar lavage (BAL) while those with HIV but no pneumonia and others with pneumonia but no HIV underwent induced sputum sampling. The collection methods have different sensitivities, with higher rates of recovery of potential pathogens using BAL sampling. In addition, there is no mention of attempts at respiratory virus sampling of the upper airway of HIV patients with pneumonia to assess whether these viruses could have been carried into the lower airways during the process of introducing the bronchoscopes.

Answer: Thanks for your comment and sorry that we missed a semicolon and the sentence was confusing. We reworded it for clarity and expanded the procedures and added the references that we followed.

We agree that different collection methods have different sensitivities, this is a challenge of pneumonia studies in immunocompetent and immunocompromised people.

Regarding your comments about the sampling we took, we collected induced sputum in all participants from the three groups following previous research from our group (Rueda ZV, López L, Marín D, Vélez LA, Arbeláez MP. Sputum induction is a safe procedure to use in prisoners and MGIT is the best culture method to diagnose tuberculosis in prisons: a cohort study. Int J Infect Dis. 2015 Apr;33:82-8. doi: 10.1016/j.ijid.2015.01.004. and Rueda ZV, Bermúdez M, Restrepo A, Garcés C, Morales O, Roya-Pabón C, Carmona LF, Arango C, Albarracín JL, López L, Aguilar Y, Maya MA, Trujillo M, Copete ÁR, Vera C, Herrera M, Giraldo MR, Niño-Cruz GI, Vélez LA. Induced sputum as an adequate clinical specimen for the etiological diagnosis of community-acquired pneumonia (CAP) in children and adolescents. Int J Infect Dis. 2022 Mar;116:348-354. doi: 10.1016/j.ijid.2022.01.026).

The BAL is protocolized and follows previously published methods by others and by our group (Vélez L, Correa LT, Maya MA, Mejía P, Ortega J, Bedoya V, Ortega H. Diagnostic accuracy of bronchoalveolar lavage samples in immunosuppressed patients with suspected pneumonia: analysis of a protocol. Respir Med. 2007 Oct;101(10):2160-7. doi: 10.1016/j.rmed.2007.05.017.).

In addition, per protocol, in these hospitals all people living with HIV who require hospitalization and are diagnosed with pneumonia or with respiratory symptoms undergo BAL. The BAL was performed with protected bronchoalveolar lavage utilizing a protective catheter to improve diagnostic yield and decrease contamination as we previously standardized (Respiratory Med 2007;101(10):2160-7. doi: 10.1016/j.rmed.2007.05.017). This protective catheter is used to avoid contamination from the upper respiratory tract pathogens. Since 2007 this procedure has been the standard in these hospitals.

The group of pneumonia without HIV, and the group with HIV without pneumonia and any respiratory symptoms, had no clinical indication for BAL, unless the treating physician considered doing so.

  1. There should be some mention of the indications for BAL in HIV patients with pneumonia at the authors' various medical centers. Do they perform BALs on all such patients? If not, this suggests that these are sicker HIV patients with pneumonia as compared to well-controlled HIV patients with a common community-acquired pneumonia (CAP). Looking at Table 1, these patients (HIV with pneumonia) have a much lower median CD4+ cell count, higher HIV RNA viral loads, and fewer were on antiretroviral therapy.

Answer: Thanks for this question and comment. As we mentioned in the previous comment, in the hospitals where we recruited participants, per protocol, all people diagnosed with HIV and pneumonia undergo BAL.

  1. The 3 subgroups of patients in this study are very different from each other in many ways (e.g., age, COPD, congestive heart failure). The comparison between them is only useful if one concludes that the recovery of respiratory viruses from any of these groups has the same lack of short or long-term impact on their illnesses.

Answer: That is correct, the groups are distinct and the degree of immunosuppression is greater in the group with HIV and pneumonia. These clinical conditions have different risk factors, for example, HIV is more common in people younger than 50 years old, and pneumonia is more common in people older than 50 years old. The study is not intended to compare similar groups but rather to assess the presence of respiratory viruses and their contribution to disease and lung dysfunction in three groups: group 1: HIV and pneumonia, group 2: HIV, and group 3: pneumonia.

  1. When was this study conducted? There is no mention of that. Curiously, looking at lines 175-187, there is no mention of looking for SARS-CoV-2 RNA.

Answer: Thanks, our apologies for missing the study period. The study was conducted between September 2016 to December 2018. We included the dates in the paper. The study predates the pandemic of and this is the reason we did not test for SARS-CoV-2.

  1. The authors should mention that, while the multiplex PCR assay utilized in this study is quite broad, it does not cover all conceivable viral causes of respiratory tract infections. Hence, the conclusions are limited to the detectable viruses.

Answer: We agree, the PCR that we used tested 16 respiratory viruses and the conclusions are limited to those viruses. We included in the methods all respiratory viruses and sub-types that panels 1, 2, and 3 of Allplex™ detect.

  1. Under Results, Figure 1, the flow chart should contain the number of patients in each group (HIV and CAP, HIV, and CAP).

Answer: Thank you, we added the number for each group.

  1. Line 239: The authors mention influenza and pneumococcal vaccine rates of uptake in the various groups, but there is no mention of any COVID-19 vaccine rates. This suggests that the study is not recent, when one takes into account the apparent lack of SARS-CoV-2 RNA testing. If so, the relevance of their data to current disease epidemiology in the various study subgroups diminishes considerably.

Answer: Thanks. We included the dates of the study. We consider that the study is relevant for all groups of participants.

  1. In Table 3, at least half the patients in each subgroup were lost to follow up. The authors should separately analyze changes in spirometric findings among those with testing at 0, 1, and 6 months for some homogeneity.

Answer: Tables 3, 4, and 5, and figure 2 report the spirometric findings at 0, 1, 6 and 12 months. The testing for respiratory viruses was done at baseline. We clarified these points in the methods. We also included details in the results about the decrease in the numbers, as some of them were deaths or withdrawals from the study.

  1. For the patients with TB, were they treated? Given the frequency of this diagnosis, perhaps an analysis of this group of patients is also warranted, since that (i.e., TB treatment) can account for observed improvement.

Answer: Yes, all participants with any treatable diagnoses were treated accordingly, including tuberculosis. We performed this analysis, but we did not find any differences between people diagnosed with tuberculosis, tuberculosis and respiratory viruses, respiratory viruses, and no tuberculosis and no respiratory viruses.

Spirometic paramenter

No respiratory virus and no TB

Respiratory virus

Respiratory virus and TB coinfection

TB

p-value Kruskal-Wallis.

Follow-up

(N=23)

(N=26)

(N=5)

(N=9)

FEV1

0

3.0 (1.7 - 3.5)

2.1 (1.8 - 3.8)

1.6 (0.7 - 3.3)

2.2 (1.3 - 2.5)

0.36

FEV1

1

3.3 (2.6 - 3.9)

2.5 (2.0 - 3.8)

2.8 (1.2 - 3.0)

2.2 (1.5 - 2.9)

0.18

FEV1

6

3.1 (2.7 - 4.0)

2.5 (2.1 - 3.9)

3.1 (3.0 - 3.3)

2.4 (1.7 - 3.0)

0.42

FEV1

12

3.3 (2.8 - 4.2)

2.9 (2.0 - 4.0)

3.4 (3.4 - 3.4)

2.5 (1.9 - 3.1)

0.54

Friedman p-value

0.427

0.520

It couldn’t be estimated due to the low number

0.112

FEV1_FVC

0

0.8 (0.8 - 0.9)

0.8 (0.7 - 0.8)

0.8 (0.7 - 0.9)

0.8 (0.7 - 0.8)

0.11

FEV1_FVC

1

0.9 (0.8 - 0.9)

0.8 (0.8 - 0.9)

0.8 (0.7 - 0.8)

0.8 (0.7 - 0.8)

0.09

FEV1_FVC

6

0.8 (0.8 - 0.9)

0.8 (0.7 - 0.9)

0.8 (0.7 - 0.9)

0.8 (0.8 - 0.8)

0.66

FEV1_FVC

12

0.8 (0.8 - 0.9)

0.8 (0.7 - 0.8)

0.9 (0.9 - 0.9)

0.8 (0.8 - 0.8)

0.24

Friedman p-value

0.043

0.250

It couldn’t be estimated due to the low number

0.615

  1. The authors should mention somewhere in the paper that patients with immune deficiency are likely to shed viruses for longer periods than immunocompetent patients. Since the PCR test is qualitative (present vs absent), the detection of any one virus does not tell us when the infection actually occurred (now or a few weeks before). Quantitative PCR assays can help, as the authors allude to in the text

Answer: Thanks. We agree with your comment. We expanded this limitation to include your suggestion: “As in other studies, defining whether the presence of respiratory viruses corresponds to asymptomatic infection ("colonization") is complex, and will continue to be a challenge that should be evaluated in future studies, perhaps by analyzing the quantitative values of RT-PCR and using repeated measures to evaluate respiratory viruses shedding after an acute infection, in particular among people with immunosuppressive conditions. Previous research has shown that certain conditions may delay PCR clearance such as older age, multiple comorbidities, and solid organ transplant but not by the degree of immunosuppression [43]. Others have shown long-term shedding of influenza, parainfluenza, and respiratory syncytial viruses in people with hematological disorders [44].”

Reviewer 2 Report

Comments and Suggestions for Authors

1. One limitation of this study is the small size of study populations, which was not discussed by the authors. Though the authors classified a total of 65 enrolled cases into three groups, ie, HIV/pneumonia (N= 27), HIV (N= 31), and pneumonia (N= 7). The small number of cases may affect the generalization of conclusions to a bigger population. 

2. The authors should add discussion that positive Pneumocystis jirovecii by PCR may not be true infection as Pneumocystis jirovecii PCR is so sensitive that it could detect Pneumocystis jirovecii colonization from asymptomatic cases. The authors are suggested to cite the following reference to facilitate this discussion. 

Development and Evaluation of a Fully Automated Molecular Assay Targeting the Mitochondrial Small Subunit rRNA Gene for the Detection of Pneumocystis jirovecii in Bronchoalveolar Lavage Fluid Specimens. J Mol Diagn. 2020 Dec;22(12):1482-1493. doi: 10.1016/j.jmoldx.2020.10.003. Epub 2020 Oct 15. PMID: 33069878.

3. Methods: there is no information on the Pneumocystis jirovecii PCR

4. It is interesting that the authors found the most frequent respiratory virus was rhinovirus (24/65, 37%). How confident was it that the PCR assay the authors used can differentiate enteroviruses from rhinoviruses? Rhinoviruses typically reside in upper respiratory tract rather than lower respiratory tract.

5. What was the season when the authors performed this study? This information should be included in the main text.

Author Response

Dear reviewer,

We want to thank you because thanks to your comments we realized that we were missing some important information, and some sentences or paragraphs were not detailed or were confusing. We incorporated the suggested changes using track changes to facilitate the review.

  1. One limitation of this study is the small size of study populations, which was not discussed by the authors. Though the authors classified a total of 65 enrolled cases into three groups, ie, HIV/pneumonia (N= 27), HIV (N= 31), and pneumonia (N= 7). The small number of cases may affect the generalization of conclusions to a bigger population. 

Answer: Thanks, we included in the paper the limitation regarding the small sample size. We agree and we do not aim to generalize the results to all population living with HIV, but this paper aims to highlight that people living with HIV and pneumonia have a high proportion of respiratory viruses that should be study in further studies. Usually, clinicians do not frequently think in respiratory viruses in people living with HIV and pneumonia.

  1. The authors should add discussion that positive Pneumocystis jirovecii by PCR may not be true infection as Pneumocystis jirovecii PCR is so sensitive that it could detect Pneumocystis jirovecii colonization from asymptomatic cases. The authors are suggested to cite the following reference to facilitate this discussion. 

 Development and Evaluation of a Fully Automated Molecular Assay Targeting the Mitochondrial Small Subunit rRNA Gene for the Detection of Pneumocystis jirovecii in Bronchoalveolar Lavage Fluid Specimens. J Mol Diagn. 2020 Dec;22(12):1482-1493. doi: 10.1016/j.jmoldx.2020.10.003. Epub 2020 Oct 15. PMID: 33069878.

 Answer: Thanks. All Pneumocystis jirovecii were diagnosed by direct stain, not by PCR. We added the references of the protocols we used.

  1. Methods: there is no information on the Pneumocystis jirovecii PCR

 Answer: We did not do Pneumocystis jirovecii PCR in our study, only stains in the BAL following previously published protocols by our group. 

  1. It is interesting that the authors found the most frequent respiratory virus was rhinovirus (24/65, 37%). How confident was it that the PCR assay the authors used can differentiate enteroviruses from rhinoviruses? Rhinoviruses typically reside in upper respiratory tract rather than lower respiratory tract.

 Answer: Thanks for your question and comment, We used a RT-PCR widely used in the literature. We agree that there are several considerations with the detection of respiratory viruses, and that the assay could not further delineate the sero-/genotypes nor differentiate between the three main species of Rhinovirus.

We used the one-step real-time RT-PCR Allplex™ Respiratory Panel 1, 2 and 3 (Seegene®, South Korea), which can detect the follow virus: 

Allplex™ Respiratory Panel 1: Influenza A virus (Flu A), Influenza A-H1 (Flu A-H1), Influenza A-H1pdm09 (Flu A-H1pdm09), Influenza A-H3 (Flu A-H3), Influenza B virus (Flu B), Respiratory syncytial virus A (RSV A), Respiratory syncytial virus B (RSV B), Internal Control (IC).

Allplex™Respiratory Panel 2: Adenovirus (AdV), Enterovirus (HEV), Metapneumovirus (MPV), Parainfluenza virus 1 (PIV 1), Parainfluenza virus 2 (PIV 2), Parainfluenza virus 3 (PIV 3), Parainfluenza virus 4 (PIV 4), Internal Control (IC) 

Allplex™ Respiratory Panel 3: Bocavirus 1/2/3/4 (HBoV), Coronavirus 229E (229E), Coronavirus NL63 (NL63), Coronavirus OC43 (OC43), Human rhinovirus (HRV), Internal Control (IC).

Other studies had evaluated Allplex™ panel 1, 2 and 3 and found concordant results with other commercial kits (Refs: Huh HJ, Kim JY, Kwon HJ, Yun SA, Lee MK, Lee NY, Kim JW, Ki CS. Performance Evaluation of Allplex Respiratory Panels 1, 2, and 3 for Detection of Respiratory Viruses and Influenza A Virus Subtypes. J Clin Microbiol. 2017 Feb;55(2):479-484. doi: 10.1128/JCM.02045-16. Epub 2016 Nov 30. 

Regarding your question about Rhinovirus, other studies have detected this microorganism in patients diagnosed with pneumonia (Walker E, Ison MG. Respiratory viral infections among hospitalized adults: experience of a single tertiary healthcare hospital. Influenza Other Respir Viruses. 2014 May;8(3):282-92. doi: 10.1111/irv.12237) and is very common in children and in patients with COPD. The frequency varies by age and comorbidities, but it has been reported as the most frequent in people living with HIV admitted to the intensive care unit with respiratory failure (Elabbadi A, Pichon J, Visseaux B, Schnuriger A, Bouadma L, Philippot Q, Patrier J, Labbé V, Ruckly S, Fartoukh M, Timsit JF, Voiriot G. Respiratory virus-associated infections in HIV-infected adults admitted to the intensive care unit for acute respiratory failure: a 6-year bicenter retrospective study (HIV-VIR study). Ann Intensive Care. 2020 Sep 14;10:123. doi: 10.1186/s13613-020-00738-9).

  1. What was the season when the authors performed this study? This information should be included in the main text.

Answer: Thanks, Colombia is a tropical country and the weather is quite similar all year around. We do not have seasons other than rainy months and warmer months. We added the dates that the study was conducted.

Round 2

Reviewer 1 Report

Comments and Suggestions for Authors

The paper is now acceptable for publication from my standpoint.

Comments on the Quality of English Language  

Good overall.

Reviewer 2 Report

Comments and Suggestions for Authors

The authors did not satisfactorily address the reviewer's comments. The major drawbacks of the study, e.g. small size of study population, compromise the quality of this study. It is too premature to published in this journal.

Comments on the Quality of English Language

NA